# Hyperuricemia and associated factors among adult cardiovascular disease patients at Salale University Comprehensive Specialized Hospital, Fitche, Central Ethiopia

Negesse Bokona Rufe[1]*, Tolera Ambisa Lamesa[2], Aklilu Getachew Mamo[2],
Belay Merkeb Zewude[1], Bedasa Addisu[3], Deresa Jamma Nigusie[1],
Sintayehu Asaye Biya[2]

1 Department of Medical Laboratory Science, College of Health Science, Salale University, Fitche, Ethiopia,
2 School of Medical Laboratory Science, Faulty of Health Science, Institute of Health, Jimma University, Jimma,
Ethiopia, 3 Department of Medical Laboratory Science, Debre Berhan University, Debre Berhan, Ethiopia

* bokonagadaa@gmail.com

## Abstract

### Background

Despite evidence suggesting that hyperuricemia (serum uric acid >7.0 mg/dL in males and >6.0 mg/dL in females) contributes to adverse outcomes like mortality and hospitalization in patients with cardiovascular disease (CVD), a comprehensive understanding of its magnitude is still lacking, underscoring the need for further investigation. There is no previously published study about hyperuricemia in cardiovascular disease in Ethiopia. Therefore, this study aims to investigate the prevalence of hyperuricemia and its associated factors among adult patients with cardiovascular diseases at Salale University Compressive Specialized Hospital Located in Fitche, Ethiopia (115 km north of Addis Ababa),from October 1, 2023, to January 28, 2024.

### Materials and methods

A hospital-based cross-sectional study was conducted on 298 participants with different types of cardiovascular disease. The participants were selected using a consecutive sampling technique. Socio-demographic factors were collected using interviewer-administered questionnaires, while overnight blood samples were collected and biochemical tests were analyzed using the COBAS c 311 automated clinical chemistry analyzer. Descriptive statistics and logistic regression analyses were performed. A variable that had a p-value of ≤ 0.05 with a 95% confidence interval was considered statistically significant.

### Result

The prevalence of hyperuricemia among adult cardiovascular disease patients was 41.3% CI [35.6–47.1]. The highest prevalence of hyperuricemia was found among

**Data availability statement:** All relevant data are within the paper and its Supporting Information files.

**Funding:** Salale University for 25000 ethiopian birr. The funders had no role in the study design, data collection and analysis, decision to publish, or preparation of the manuscript.

**Competing interests:** The authors have declared that no competing interests exist.

cardiovascular disease patients with congestive heart failure (48.0%) and hypertensive heart disease (41.2%). Factors such as physical activity (AOR: 4.1; 95% CI: 1.5–10.7, P = 0.004), dyslipidemia (AOR: 2.7; 95% CI: 1.2–6.0, P = 0.01) and chronic kidney disease (AOR: 3.1; 95% CI: 1.5–6.1, P = 0.001) were found to be associated with hyperuricemia among individuals with cardiovascular disease.

## Conclusion

The study indicated a high prevalence of hyperuricemia among participants with cardiovascular disease. Physical activity, dyslipidemia, and chronic kidney disease were recognized as significant associated factors of hyperuricemia among cardiovascular disease. Therefore, early diagnosis of hyperuricemia and its management is essential to control complications and extend the life expectancy of individuals with cardiovascular disease.

## Introduction

Cardiovascular diseases (CVDs) represent a category of non-communicable diseases that arise from damage to the heart and blood vessels, impacting the cardiovascular system along with its associated structures [1]. CVDs represent a diverse group of conditions affecting the heart and blood vessels. These include congestive heart failure (CHF), a state in which the heart's pumping capacity is insufficient to meet the body's metabolic needs, and hypertensive heart disease (HHD), which refers to cardiac dysfunction resulting from chronic hypertension. Cardiomyopathy (CMP) is another CVD characterized by abnormalities in the heart muscle, such as enlargement, thickening, or rigidity. Valvular heart disease (VHD) arises from damage to the heart's valves and/or muscles, impairing its function. Rheumatic heart disease (RHD) is a consequence of rheumatic fever, leading to cardiac injury. Finally, stroke, a critical cerebrovascular event, encompasses conditions such as thrombus formation in atherosclerotic cerebral blood vessels and small vessel disease within the brain [2–5]. It is one of the primary contributors to global mortality, its prevalence fluctuates significantly, ranging from 4% in high-income nations to 42% in low-income nations [6,7].

Uric acid is the final product of purine catabolism originating from various body organs such as the liver, gut, vascular endothelium, kidney, and myocardium cells. Additionally, it is produced from damaged cells through the action of the xanthine oxidase enzyme. The production and excretion of serum uric acid (SUA) are always kept in balance [8]. Overproduction or under-excretion of SUA can lead to its increase in blood and finally cause hyperuricemic metabolic disorder which is defined as serum uric acid ≥ 7 mg/dl for males and ≥ 6 mg/dl for females [9,10].

Hyperuricemia is a metabolic disorder that is prevalent worldwide and is linked to a broader range of non-communicable diseases such as CVD, Diabetes Mellitus (DM), and hypertension, with a prevalence ranging from 2.3% to 26%. Hyperuricemia is the fourth highest metabolic disorder after hypertension, hyperglycemia, and hyperlipidemia due to increasingly unhealthy lifestyles [11].

Hyperuricemia can induce vascular damage by promoting oxidative stress, impairing nitric oxide production, and causing endothelial dysfunction. These effects, coupled with local and systemic inflammation, contribute to vasoconstriction, vascular smooth muscle proliferation, atherosclerosis, and activation of the renin-angiotensin-aldosterone system, ultimately impacting cardiovascular health [12–14].

Research indicates that hyperuricemia serve as an independent risk factor for a variety of cardiovascular diseases, including hypertension, coronary artery disease, heart failure, and stroke. An increase in serum uric acid concentration disrupts the equilibrium of antioxidants, thereby facilitating the onset of cardiovascular disease [11,15].

The European Society of Cardiology (ESC) and the European Society of Hypertension (ESH) have recently acknowledged uric acid as a comorbidity associated with cardiovascular disease [16]. Furthermore, various research findings have demonstrated that each 1 mg/dl increase in hyperuricemia is linked to heightened mortality from cardiovascular disease in both older males and females [17,18]. Some studies also showed that between 30% and 60% of individuals experiencing either acute or chronic heart failure exhibit hyperuricemia. Furthermore, research reveals that 23.9% of adults in the general Spanish population diagnosed with cardiovascular disease present elevated UA, along with 46.5% of patients with coronary artery disease (CAD) in South India, and 37.4% of patients with CAD and arterial hypertension in Switzerland [19,20].

Limited research from Africa suggests a link between hyperuricemia and cardiovascular disease (CVD). A 2021 Libyan study by Ali Fadhlullah et al. found that 69.1% of individuals with hyperuricemia also had coronary artery disease (CAD) [21], supporting this association. However, this retrospective, single-center study had a small sample size, introducing potential biases and limiting its generalizability, particularly regarding the prevalence of hyperuricemia across diverse CVD subtypes.

Ethiopian studies by Bedesa et al. (2023) and Abebe Timerga et al. (2021) reported hyperuricemia prevalence of 43.1% in cardiac patients and 27.4% in essential hypertensive patients, respectively [9,22]. These studies did not investigate hyperuricemia across the diverse subtypes of CVD and extensively examine potential associated factors. Various factors, including age, alcohol consumption, smoking behaviors, and body weight, significantly contribute to the onset of hyperuricemia in individuals afflicted with (CVDs). Furthermore, hyperuricemia is notably associated with the emergence of comorbid conditions related to cardiovascular diseases, including dyslipidemia, diabetes mellitus, and hypertension, thereby exacerbating the complexities of CVDs [23–25].

Ethiopia faces a significant public health challenge with its high burden of cardiovascular disease (CVD), a condition for which hyperuricemia is a known global risk factor. Despite international guidelines and consensus recommending screening and treatment for hyperuricemia in cardiovascular disease (CVD), supported by numerous studies [11,16,26], there is a notable lack of data on hyperuricemia impact on CVD in Africa, including Ethiopia. Furthermore, there are currently no published studies investigating this relationship specifically within the Ethiopian context. This hinders the development of effective prevention and management strategies. Therefore; our research aims to fill this gap by comprehensively assessing hyperuricemia across various CVD subtypes in Ethiopia. The findings will inform improved clinical guidelines and public health initiatives to reduce CVD burden and enhance patient outcomes in Ethiopia and similar African contexts.

## Materials and methods

### Study area, design and period

The study was conducted at Salale University Comprehensive Specialized Hospital, which is the only comprehensive specialized hospital in the North Shewa Zone of Oromia Regional State. The hospital is located in Fitche town, 115 km north of Addis Ababa, the capital city of Ethiopia. Currently, the hospital has 329 health professionals, 24 specialists and serves over 1.6 million people in the catchment area. Hence, a hospital-based cross-sectional study was conducted on adult patients diagnosed with cardiovascular diseases attending the hospital to assess metabolic syndrome from October 1, 2023, to January 28, 2024.

## Sample size and sampling technique

Sample size was calculated using the single population proportion formula taken from the study conducted in Ambo, Ethiopia (p = 43.1%) [9], using a 95% Confidence Interval and a 5% degree of precision. Since the source population (964) was less than 10,000, the correction formula was used and a 10% non-response rate was added. Finally, 298 study participants were recruited in the study using consecutive sampling technique.

## Eligibility criteria

We included adult patients diagnosed with cardiovascular disease and willing to participate in the study during the study period. Participants who were taking medication that lowers uric acid and lipid profile, those with serious illness and could not provide information, and chemotherapy users were excluded from the study.

## Data collection procedure

Interviewer-administered questionnaires were used to collect sociodemographic, clinical, and lifestyle-related factors, as well as anthropometric data. Trained BSc Nurses collected sociodemographic, lifestyle, and anthropometric data. Clinical data was collected by physicians who followed cardiovascular disease patients. Various types of cardiovascular disease and the length of medication were determined by reviewing the patients' medical records. The presence of comorbidities such as diabetes, dyslipidemia, and chronic kidney diseases was established through corresponding biochemical analysis and physician clinical discussion.

## Anthropometric measurements

Anthropometric measurements such as body mass index (BMI) and central obesity were taken by a clinical nurse. The participant wore light-closed shoes and stood on a scale on a level floor. Body weight and height were also measured using a Conxport height measure with a mechanical weighing scale manufactured in Ambala Cantt, Haryana, India in 1997. BMI was calculated based on the World Health Organization (WHO) guidelines from 2023, which is weight (kg) divided by height ($m^2$) squared. Body weight categories were defined based on BMI as underweight (<18.5 kg/$m^2$), normal (18.5–24.99 kg/$m^2$), overweight (25–29.9 kg/$m^2$), and obesity (>30 kg/$m^2$) [27]. For central obesity, waist circumference was measured at the level of the iliac crest using a non-elastic tape measurement from Jig Pro Shop LLC in the United States. Central obesity is defined as a waist circumference greater than 94 cm for males and greater than 80 cm for females [8].

## Laboratory measurements

After getting consent from the study participant, 5 ml of fasting venous blood was collected with a serum separator tube. After 30–45 minutes of clotting and centrifugation, serum was separated and analyzed for determination of lipid profile, uric acid, and creatinine. The appropriate sample and specific reagents are combined and incubated to initiate a chemical reaction. Biomolecules such as Uric acid, lipid profile (cholesterol, triglycerides HDL-c, LDL-c), and serum glucose are analyzed using enzymatic colorimetric assay whereas creatinine is determined by kinetic colorimetric assay [28].

   Clinical factors was identified using these biomarkers and clinician judgment. Hyperuricemia occurs when uric acid level ≥ 7 mg/dl in men and ≥ 6 mg/dl in women [9,10]. Diabetes mellitus is diagnosed when a person's fasting serum glucose is ≥ 126 mg/dl [29]. Dyslipidemia is defined by the National Cholesterol Education Program Adult Panel III guidelines when total cholesterol (TC) ≥ 200 mg/dl, HDL cholesterol (HDL-c) <40 mg/dl for men and < 50 mg/dl for women, LDL cholesterol (LDL-c) ≥ 130 mg/dl and triglycerides (TG) ≥ 150 mg/dl [30]. Chronic kidney disease is identified when the estimated glomerular filtration rate (eGFR) is <60 mL/min per 1.73 m² for over three months, calculated using the CKD-EPI formula [31]. For analysis of selected biomarkers, COBAS c311 automated clinical chemistry analyzer (Hitachi High-Technologies Corporation, Tokyo, Japan) was used.

## Operational definitions

**Alcohol drinkers.**  Current drinkers are those aged 18 and older who have consumed alcohol in the past year. Non-alcohol drinkers have never consumed alcohol, while former drinkers are those who used to drink but have not in the last year [32].

**Cigarette smokers.**  Current smokers are individuals who have smoked in the past year, whereas non-smokers have never smoked or have not smoked in the last year [8].

**Khat chewers.**  Current chewers are those who have chewed khat in the past year, while non-chewers have never chewed it or have not done so in the last year [8].

**Physical activity.**  Low-level physical activity refers to engaging in any physical activity, like walking, for less than 30 minutes on five days a week. Moderate physical activity involves activities that raise the heart rate for 150–300 minutes each week. Vigorous physical activity includes intense activities like heavy lifting or running for 75–150 minutes weekly [33].

## Data processing and analysis

The data were entered into Epi Data version 4.6 and analyzed using Statistical Package for Social Science (SPSS) version 26 (SPSS, Chicago, USA) statistical software. Descriptive analyses such as cross-tabulation and frequency were used to explain the distribution of variables associated with hyperuricemia. The odds ratio and 95% confidence interval of logistic regressions were used to measure the strength of an association between hyperuricemia and its associated factors among CVDs. Initially, we reviewed the assumptions and evaluated the model's fit. Then, we examined the relationship between each independent variable and hyperuricemia using binary logistic regression. All variables whose p-values were <0.25 by binary logistic regression analysis were selected for multiple logistic regression for the final selection of associated factors. After multiple logistic regression, all variables whose p-value ≤ 0.05 and AOR > 1 were recognized as final associated factors for hyperuricemia in cardiovascular disease. The assumption, of logistic regression was checked by Spearman correlation coefficient and collinearity diagnosis such as (Tolerance value >0.1 and VIF (Variance Inflation Factor) <10). Again for goodness of fit model (Omnibus test of model p-value ≤0.05 and Hosmer and Lemeshow p-value ≥0.05) were also checked (S2 Table).

## Data quality management

The quality of the data was regularly followed throughout every procedure. For the validity of questionnaires, a pre-test was conducted on 5% of the sample size of study participants at Muke Turi Primary Hospital before collecting actual data. The questionnaires were translated from the English version to the local language for consistency and clarity. A blood sample was collected, stored, and transported according to standard operating procedures. The calibrator, quality control, and test reagent expiration dates were inspected before starting the analysis procedure. The startup procedure, maintenance, calibration, and internal quality control were done according to job aids and standard operating procedures of the analyzer before running participant samples. All requisite procedures and protocols were adhered to in accordance with the manufacturer's guidelines. The principal investigator conducted daily assessments to ensure the completeness of the collected results. Patients with abnormal results were linked to a physician.

## Ethical approval and consent to participate

Ethical clearance Letter was obtained from the institutional review board of the Institute of Health Science, Jimma University, and Ref. No: JUIH/IRB/585/23. The purpose, benefit, and method of study were clearly explained to the participants. Each of the participants was assured that their responses would remain confidential. Written informed consent was obtained from the participants before enrollment in the study, and those willing to participate were included. To ensure

confidentiality, study participants were identified by code, and only authorized persons could collect and access the data. The collected specimen was used only for the stated objective.

## Result

### The distribution of socio-demographic and lifestyle factors among participants with CVDs

The study included 298 participants with different types of cardiovascular disease. The majority of participants were female, accounting for 58.4%, and the larger number of participants (54%) were those who were aged ≥55 years. Fifty-five percent (55%) of participants were from rural areas, whereas a larger number of the participants were house-wives, accounting for 36.6%, while 62.8% of individuals were unable to read and write. Only a small percentage of the participants, 4.7%, were smokers, and the majority (36.6%) of them were current alcohol drinkers. The percentage of participants who were current khat chewers and had low physical activity was 1.7% and 44.3%, respectively (Table 1).

### Clinical and biochemical characteristics among participants with CVDs

From the total study participants, the largest number of individuals were diagnosed with hypertensive heart disease (38.3%) and congestive heart failure (33.6%). Individuals who developed valvular heart disease, cardiomyopathy, rheumatic heart disease, and stroke accounted for 9.4%, 8.7%, 5.7%, and 4.4%, respectively. Forty-four percent (44%) of participants had a one to five-year duration of CVD. This study showed that the participants with fasting serum blood

Table 1. The distribution of sociodemographic characteristics and lifestyle factors of study participants (n = 298).

| Parameters | Categories | Number (298) | Percent (%) |
|---|---|---|---|
| Sex | Male | 124 | 41.6 |
| | Female | 174 | 58.4 |
| Age (in years) | ≥55 | 165 | 55.4 |
| | <55 | 133 | 44.6 |
| Residence | Rural | 164 | 55 |
| | Urban | 134 | 45 |
| Occupation | Housewife | 109 | 36.6 |
| | Farmer | 70 | 23.5 |
| | Employed | 36 | 12.1 |
| | Private worker | 77 | 25.8 |
| | Student | 6 | 2 |
| Education | Unable to write and read | 187 | 62.8 |
| | Able to write and read | 36 | 12.1 |
| | Primary school | 32 | 10.7 |
| | Secondary school | 21 | 7 |
| | College and above | 22 | 7.4 |
| Cigarette smoking | Yes | 14 | 4.7 |
| | No | 284 | 95.3 |
| Alcohol consumption | Current drinkers | 109 | 36.6 |
| | Former drinkers | 77 | 25.8 |
| | Non-drinkers | 112 | 37.6 |
| Khat chewing | Yes | 5 | 1.7 |
| | No | 293 | 98.3 |
| Physical activity | Low | 132 | 44.3 |
| | Moderate | 102 | 34.2 |
| | High | 64 | 21.5 |

glucose ≥ 126 accounted for 10.4%, while those with a lipid profile (total cholesterol ≥200 mg/dl, LDL > 130 mg/dl, and tri-glyceride >150 mg/dl) were 28.6%, 35.9%, 29.7%, and 32.9%, respectively. On the other side, HDL < 40 mg/dl (male) and <50 mg/dl (female) were accounted for 25.2% and 24.8%, respectively. The number of males with serum uric acid ≥7 mg/dl and females with serum uric acid ≥6 mg/dl were 17.8% and 23.5%, respectively (Table 2).

During the study period, participants with central obesity and chronic kidney disease accounted for 29.9% and 28.9% of the total population, respectively. In addition, 23.5% and 10.4% of participants developed dyslipidemia and diabetes mellitus, while 41.3% developed hyperuricemia. The majority of participants 52.7% of the group, were outpatients undergoing follow-up for cardiovascular diseases (Fig 1).

### The prevalence of hyperuricemia in relation to sociodemographic, clinical, and lifestyle factors among participants with (CVDs)

Among a total of 298 CVD patients, 123 individuals developed hyperuricemia, which accounted for 41.3% (CI [35.6–47.1]). In this study, a higher prevalence of hyperuricemia was observed among females (42.7%) compared to males. Additionally, individuals aged over the average of 55 years had a higher prevalence of hyperuricemia (52.1%). The higher magnitude of hyperuricemia (45.2%) and (45.9%) was found among CVD patients living in rural areas and among

**Table 2. Clinical and biochemical characteristics among participants with cardiovascular diseases (n = 298).**

| Variables | Category | | Numbers (%) |
|---|---|---|---|
| CVDs types | HHD | | 114 (38.3) |
| | CHF | | 100 (33.6) |
| | RHD | | 17 (5.7) |
| | VHD | | 28 (9.4) |
| | Stroke | | 13 (4.4) |
| | CMP | | 26 (8.7) |
| Duration of CVDs | >10 years | | 42 (14.1) |
| | 6-10 years | | 85 (28.5) |
| | 1-5 years | | 131 (44) |
| | <1 year | | 40 (13.4) |
| BMI status | BMI ≥ 30 kg/m² | | 11 (3.7) |
| | BMI 25–29.9 kg/m² | | 79 (26.5) |
| | BMI 18.5–24.99 kg/m² | | 148 (49.7) |
| | BMI < 18.5 kg/m² | | 60 (20.1) |
| Total cholesterol | ≥ 200 mg/dl | | 81 (27.2) |
| | <200mg/dl | | 217 (72.8) |
| HDL-c | Male | < 40 mg/dl | 75 (25.2) |
| | | ≥40 mg/dl | 49 (16.4) |
| | Female | < 50 mg/dl | 74 (24.8) |
| | | ≥50 mg/dl | 100 (33.6) |
| LDL-c | ≥ 130 mg/dl | | 88 (29.7) |
| | <130 mg/dl | | 210 (70.3) |
| Triglyceride | ≥ 150 mg/dl | | 98 (32.9) |
| | <150 mg/dl | | 200 (67.1) |
| Serum uric acid | Male | ≥7 mg/dl | 53 (17.8) |
| | | < 7 mg/dl | 71 (23.8) |
| | Female | ≥6 mg/dl | 70 (23.5) |
| | | <6 mg/dl | 104 (34.9) |

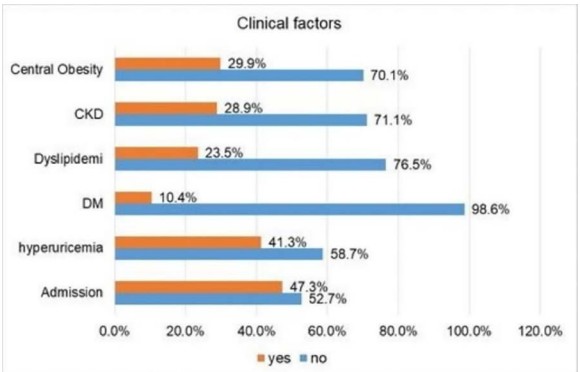

**Fig 1. Clinical factors among participants with cardiovascular diseases at SUCSH from October 1, 2023– January 28, 2024 (n = 298).**

housewives, respectively. The present findings revealed that hyperuricemia was present in 63.4% of cigarette smokers, whereas among current alcohol drinkers, it was 63.3%. Only two individuals who chewed Khat developed hyperuricemia. Moreover, participants with low physical activity had the highest prevalence of hyperuricemia at 60.6%, compared to those with moderate and high physical activity. The prevalence of hyperuricemia was also higher, at 64.6%, among individuals with overweight based on BMI. Among CVD patients with central obesity, diabetes, and dyslipidemia hyperuricemia was accounted for 64%, 61.3% and 68.6% respectively (Table 3).

In terms of the overall prevalence, hyperuricemia was most frequently identified in patients diagnosed with congestive heart failure (48.0%) and hypertensive heart disease (41.2%). Hyperuricemia was also noted in individuals with valvular heart disease (39.3%), cardiomyopathy (34.6%), rheumatic heart disease (30.8%), and stroke (23.5%) (Fig 2).

### Associated factors of hyperuricemia among cardiovascular disease participants

The present study identified the associated factors of hyperuricemia among participants with cardiovascular diseases (CVDs). The dependent variables, such as sex, age, residence, occupation, education, cigarette smoking, alcohol consumption, khat chewing, physical activity, body weight, central obesity, DM, dyslipidemia, CKD, and duration of CVDs, were selected as potential associated variables. However, factors such as age, residence, education, cigarette smoking, alcohol consumption, physical activity, body weight, central obesity, DM, dyslipidemia, CKD, and duration of CVDs were selected for multiple logistic regression because they had a p-value <0.25 by binary logistic regression analysis (S2 Table). After conducting multiple logistic regression analysis, factors such as physical activity (p-value = 0.004), dyslipidemia (p-value = 0.01), and chronic kidney disease (p-value = 0.001) were identified as the final associated factors of hyperuricemia among participants with CVDs at a P-value ≤ 0.05 and AOR > 1 (Table 4).

Individuals with low physical activity were about 4.1 times more likely to develop hyperuricemia than those who had regular physical activity among CVDs (AOR: 4.1; 95% CI: 1.5–10.7, P = 0.004). Hyperuricemia was about 2.7 times more likely to be developed among participants who had dyslipidemia than those who did not have (AOR: 2.7; 95% CI: 1.2–6.0, P = 0.01). Participants with chronic kidney disease had also 3.1 times higher likelihood of having hyperuricemia compared to those without chronic kidney disease (AOR: 3.1; 95% CI: 1.5–6.1, P = 0.001) (Table 4).

### Discussion

Several studies have indicated that hyperuricemia is an independent risk factor for both mortality and complications of cardiovascular disease. It is recommended that individuals with cardiovascular disease undergo screening for hyperuricemia [14,34,35]. Though the association and complications of CVDs induced by hyperuricemia have been well studied, there

**Table 3. The prevalence of hyperuricemia in relation to sociodemographic, clinical, and lifestyle factors among participants with cardiovascular diseases (n = 298).**

| Parameters | Categories | Hyperuricemia (130) | | Normouricemia (168) | | X² | P-value |
|---|---|---|---|---|---|---|---|
| | | No. | % | No. | % | | |
| Sex | Male | 53 | 42.7 | 71 | 53.7 | 1.88 | 0.66 |
| | Female | 70 | 40.2 | 104 | 50.8 | | |
| Age (in years) | ≥55 | 86 | 52.1 | 79 | 47.9 | 17.94 | <0.001 |
| | <55 | 37 | 27.8 | 96 | 72.7 | | |
| Residence | Rural | 74 | 45.1 | 90 | 45.9 | 2.22 | 0.136 |
| | Urban | 49 | 36.6 | 85 | 63.4 | | |
| Occupation | Housewife | 50 | 45.9 | 59 | 54.1 | 2.989 | 0.556 |
| | Farmer | 31 | 44.3 | 39 | 55.7 | | |
| | Employed | 13 | 36.1 | 23 | 63.9 | | |
| | Private worker | 27 | 35.1 | 50 | 64.9 | | |
| | Student | 2 | 33.3 | 4 | 66.7 | | |
| Education | Unable to write and read | 93 | 49.7 | 94 | 50.3 | 16.03 | 0.030 |
| | Able to write and read | 9 | 25 | 27 | 75 | | |
| | Primary school | 7 | 21.9 | 25 | 78.1 | | |
| | Secondary school | 6 | 28.6 | 15 | 71.4 | | |
| | College and above | 8 | 36.4 | 14 | 63.6 | | |
| Cigarette smoking | Yes | 9 | 64.3 | 5 | 35.7 | 2.550 | 0.110 |
| | No | 114 | 40.1 | 170 | 59.9 | | |
| Alcohol consumption | Current drinkers | 69 | 63.3 | 40 | 36.7 | 38.84 | <0.001 |
| | Former drinkers | 29 | 37.7 | 48 | 62.3 | | |
| | Non drinkers | 25 | 22.3 | 87 | 77.7 | | |
| Khat chewing | Yes | 2 | 40 | 3 | 60 | 0.03 | 0.869 |
| | No | 121 | 41.3 | 172 | 59.7 | | |
| Physical activity | Low | 80 | 60.6 | 52 | 39.4 | 41.072 | ≤0.001 |
| | Moderate | 33 | 32.4 | 69 | 67.6 | | |
| | High | 10 | 15.6 | 54 | 84.4 | | |
| BMI status | Obesity | 6 | 54.5 | 5 | 45.5 | 27.130 | ≤0.001 |
| | Over weight | 51 | 64.6 | 28 | 35.4 | | |
| | Normal | 44 | 29.7 | 104 | 70.3 | | |
| | Underweight | 22 | 36.7 | 38 | 63.3 | | |
| Central obesity | Yes | 57 | 64 | 32 | 36 | 27.144 | ≤0.001 |
| | No | 66 | 31.6 | 143 | 69.4 | | |
| DM | Yes | 19 | 61.3 | 12 | 38.7 | 5.714 | 0.017 |
| | No | 104 | 39 | 163 | 61 | | |
| Dyslipidemia | Yes | 48 | 68.6 | 22 | 31.4 | 28.124 | 0.01 |
| | No | 75 | 32.9 | 153 | 67.1 | | |
| CKD | Yes | 63 | 73.3 | 23 | 26.7 | 51.009 | <0.001 |
| | No | 60 | 28.3 | 152 | 71.7 | | |
| Admission status | Admitted | 85 | 60.3 | 56 | 39.7 | 39.895 | ≤0.001 |
| | Out patients | 38 | 24.2 | 119 | 75.8 | | |
| Duration of CVDs | >10 years | 22 | 52.4 | 20 | 47.6 | 39.895 | 0.01 |
| | 6-10 years | 39 | 45.9 | 46 | 54.1 | | |
| | 1-5 years | 53 | 40.5 | 78 | 59.5 | | |
| | <1 year | 9 | 22.5 | 31 | 77.5 | | |

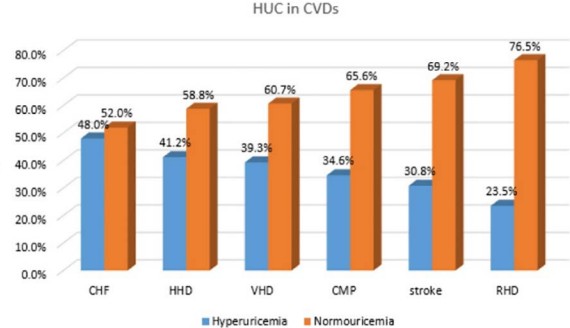

**Fig 2. Prevalence of hyperuricemia among each cardiovascular disease types of study participants at SUCSH, from October 1, 2023– January 28, 2024, (n = 298).**

is still limited information on the overall magnitude of hyperuricemia and its associated factors among CVDs in Ethiopia, including the study area.

In this study, the overall prevalence of hyperuricemia among adult patients with various cardiovascular diseases was 41.3% CI [35.6–47.1]. Our finding aligns with a previous study conducted by Addisu Bedasa et al. in 2023 in Ambo, Ethiopia, which reported 43.1% of hyperuricemia among cardiac patients [9]. The reason for the consistency was due to geographical area, study participants, and lifestyle factors.

The current finding of hyperuricemia was less than the result reported in a recent retrospective cross-sectional study by Ali Fadhlullah *et al.* in Libya in 2021, which indicated that 69.1% (67 out of 97) of hyperuricemia among individuals with coronary artery disease (CAD) [21]. The discrepancy might be attributed to the fact of sample size, geographic location and retrospective nature of data.

This result aligns with research conducted by Topolyanskaya et al. in Moscow, Russia, in 2019, which reported that 37.4% of hyperuricemia among patients presented with cardiovascular diseases, including coronary artery disease and arterial hypertension [36]. Additionally, a study by Saklesh Patil et al. in South India the same year found that 46.5% (95% CI: 42.29 to 50.84%) of hyperuricemia among individuals with CVDs [23]. The reason of consistance might be due to similarity of study design, sample size and major types of CVDs (heart failure, and stroke).

The present results is higher than the finding by Paula Antelo-Pais and colleagues in Spain in 2022, which found a hyperuricemia prevalence of 23% among patients with cardiovascular diseases (CVDs) [20]. Furthermore, two studies from China, one by Khalid Mehmood et al. in 2021 [37], and another by Yao-Wei Zou et al. in 2022 [38] reported hyperuricemia prevalence rates of 29% and 27.9%, respectively, among individuals with CVD. This discrepancy may be attributed to various factors, including the characteristics of their study participants (those with a history of CVDs and those at risk), lifestyle factors, sociodemographic variables, and the study design, all of which could influence the observed prevalence of hyperuricemia.

Purine-rich diets and genetic factors are known contributors to elevated serum uric acid levels [15,35]. However, research specifically investigating the combined influence of dietary habits and genetic predispositions on uric acid levels within the Ethiopian population is currently lacking. Further studies are needed to elucidate the specific interplay of these factors in this population.

The present finding revealed that there is an association between low physical activity and hyperuricemia (AOR: 4.1; 95% CI: 1.5–10.7, P = 0.004). This finding lines up with a study conducted by Saki *et al.* in Japan from 2005–2018, which concluded that low physical activity was linked to hyperuricemia in individuals with various cardiovascular diseases, with a significance level of p < 0.001 [39]. The reason for the agreement between the two studies might be due to individuals who

**Table 4. Binary and multiple logistic regression to identify associated factors of hyperuricemia among cardiovascular disease patients at SUCSH from October 1, 2023– January 28, 2024.**

| Variables | Category | Hyperuricemia | | COR [95%CI] | P-value | AOR [95%CI] | P-value |
|---|---|---|---|---|---|---|---|
| | | Yes | No | | | | |
| Age (in years) | ≥55 | 86 (52.1%) | 79 (47.9%) | 0.3 [0.2-0.5] | < 0.001 | 1.6 [0.8-3.1] | 0.12 |
| | <55 | 37 (27.8%) | 96 (72.7%) | 1 | | | |
| Residence | Urban | 74 (45.1%) | 90 (45.9%) | 1.4 [0.8-2.2] | 0.13 | 0.9 [0.4-1.9] | 0.95 |
| | Rural | 49 (36.6%) | 85 (63.4%) | 1 | | | |
| Education | Unable to write and read | 93 (49.7%) | 94 (50.3%) | 0.5 [0.2-1.4] | 0.24 | 1.7 [0.4-6.3] | 0.40 |
| | Able to write and read | 9 (25%) | 27 (75%) | 1.7 [0.5-5.4] | 0.35 | 0.4 [0.1-2.3] | 0.37 |
| | Primary school | 7 (21.9%) | 25 (78.1%) | 2[0.6-6.8] | 0.24 | 0.4 [0.0-2.0] | 0.27 |
| | Secondary school | 6 (28.6%) | 15 (71.4%) | 1.4 [0.4-5.1] | 0.58 | 0.4 [0.0-2.1] | 0.30 |
| | College and above | 8 (36.4%) | 14 (4.763.6%) | 1 | | | |
| Cigarette smoking | Yes | 9 (64.3%) | 5 (35.7%) | 0.4 [0.1-1.1] | 0.08 | 3.4 [0.8-14.5] | 0.08 |
| | No | 114 (40.1%) | 170 (59.9%) | 1 | | | |
| Alcohol consumption | Current drinker | 69 (63.3%) | 40 (36.7%) | 0.2 [0.1-0.3] | < 0.001 | 2.1[0.9-4.7] | 0.058 |
| | Former drinkers | 29 (37.7%) | 48 (62.3%) | 0.4[0.2-0.9] | 0.02 | 1.4 [0.6-3.4] | 0.34 |
| | Non- drinkers | 25 (22.3%) | 87 (77.7%) | 1 | | | |
| Physical activity | Low | 80 (60.6%) | 52 (39.4%) | 0.1 [0.05-0.2] | < 0.001 | 4.1 [1.5-10.7] | **0.004** |
| | Moderate | 33 (32.4%) | 69 (67.6%) | 0.3 [0.1-0.8] | 0.02 | 1.9 [0.7-4.7] | 0.18 |
| | High | 10 (15.6%) | 54 (84.4%) | 1 | | | |
| BMI | Obesity | 6 (54.5%) | 5 (45.5%) | 0.4 [0.1-1.7] | 0.27 | 0.9 [0.1-6.5] | 0.99 |
| | Over weight | 51 (64.6%) | 28 (35.4%) | 0.3 [0.1-0.6] | 0.001 | 1.6 [0.5-4.8] | 0.34 |
| | Normal | 44 (29.7%) | 104 (70.3%) | 1.3 [0.7-2.5] | 0.33 | 0.9 [0.4-2.0] | 0.85 |
| | Underweight | 22 (36.7%) | 38 (63.3%) | 1 | | | |
| Central obesity | Yes | 57 (64%) | 32 (36%) | 0.2 [0.1-0.4] | < 0.001 | 2.1[0.8-4.9] | 0.12 |
| | No | 66 (31.6%) | 143 (69.4%) | 1 | | | |
| DM | Yes | 19 (61.3%) | 12 (38.7%) | 0.4 [0.2-0.8] | 0.02 | 2.6 [0.9-7.6] | 0.06 |
| | No | 104 (39%) | 163 (61%) | 1 | | | |
| Dyslipidemia | Yes | 48 (68.6%) | 22 (31.4%) | 0.2 [0.1-0.3] | < 0.001 | 2.7 [1.5-6.0] | **0.011** |
| | No | 75 (32.9%) | 153 (67.1%) | 1 | | | |
| CKD | Yes | 63 (73.3%) | 23 (26.7%) | 0.1 [0.08-0.2] | < 0.001 | 3.1[1.5-6.1] | **0.001** |
| | No | 60 (28.3%) | 152 (71.7%) | 1 | | | |
| Duration of CVDs | >10 year | 22 (52.4%) | 20 (47.6%) | 0.2 [0.1-0.6] | 0.01 | 1.5 [0.4-5.6] | 0.47 |
| | 6-10 year | 39 (45.9%) | 46 (54.1%) | 0.43[0.2-0.8] | 0.01 | 1.9 [0.6-5.7] | 0.23 |
| | 1-5 year | 53 (40.5%) | 78 (59.5%) | 0.4 [0.1-0.9] | 0.17 | 1.5 [0.5-4.2] | 0.43 |
| | <1 year | 9 (22.5%) | 31 (77%) | 1 | | | |

COR: Crude Odd Ratio, AOR: Adjusted Odd Ratio, Ref. –Reference value, P-value <0.05 is statistically significant.

are physically inactive or low can be at risk for weight gain and metabolic disorders, which can increase the risk of hyperuricemia [40].

Additionally, this finding identified a significant association between dyslipidemia and hyperuricemia (AOR: 2.7; 95% CI: 1.2–6.0, P = 0.01). This finding is consistent with a cross-sectional study carried out by Irfan *et al.* in Pakistan in 2020, (p-value of 0.002), and seki et al. in Japan in 2021, (p-value< 0.001) reported a significant association between elevated SUA levels and dyslipidemia and its components in individuals with CVD, with a p-value of 0.002 and p-value <0.001

respectively [39,41]. This study is consistence because oxidized LDL and other lipid profiles cause endothelial dysfunction that lead to leakage of high uric acid [13].

The present study revealed that chronic kidney disease is a risk factor for hyperuricemia among cardiovascular disease (AOR: 3.1; 95% CI: 1.5–6.1, P = 0.001) This finding is also supported by the study conducted by Liu *et al.* in 2024, in Chinese, and also by Lee *et al.* in 2023 in Korea which indicated that hyperuricemia can be developed due to chronic kidney disease [42,43]. The reason for agreement between these studies is that 90% of UA is filtered through the renal and hyperuricemia can happen if the kidney is injured due to adverse effects from CVDs medication or defect of renal UA transporter genes [44]. Hyperuricemia, in the blood, develops due to either overproduction of uric acid or, more commonly, impaired excretion by the kidneys. Renal disease directly contributes to hyperuricemia because damaged kidneys are less efficient at filtering and removing uric acid from the bloodstream. This disrupted clearance process causes uric acid to accumulate [45].

In summary, our finding of a high prevalence of hyperuricemia among Ethiopian cardiovascular disease (CVD) patients, alongside the current absence of routine screening and management, indicates a significant and overlooked comorbidity likely exacerbating CVD complications and mortality. This highlights a critical gap in CVD care in Ethiopia, demanding urgent attention to the potential role of hyperuricemia in worsening cardiovascular outcomes. Implementing uric acid screening for CVD patients and raising awareness among healthcare providers are crucial first steps. While further research is needed to fully understand the specific causes and best management strategies in this population, our findings strongly suggest that addressing hyperuricemia could significantly improve CVD care and reduce related complications in Ethiopia.

### Limitation of the study

We acknowledge the inherent limitations of self-reported lifestyle data, particularly concerning recall bias in variables such as physical activity and alcohol consumption. While standardized questionnaires and clear instructions were employed to mitigate this potential bias, the possibility remains that participants may have underreported certain behaviors. Future research should consider incorporating objective measures to minimize recall bias and provide a more accurate assessment of these associations.

The study was also limited by not exploring the causal relationship between hyperuricemia and CVDs due to the cross-sectional nature of study design. We used consecutive sampling, a non-probability technique that could introduce selection bias. To mitigate this, we extended data collection over three months, aiming to increase the probability of all eligible individuals being included in the study. Furthermore, we considered that consumption of a purine-rich diet can affect serum uric acid, but we did not include it as a measurable factor because there is no established tool to assess purine-rich diet items, frequency, and even the dietary styles and items can vary among countries. Despite its limitation the study findings fills the gap of the scarcity of magnitude, pooled prevalence, and associated factors of hyperuricemia among CVDs and its subtypes from a low-income country, including Ethiopia.

### Conclusion and recommendation

In our study, nearly half of individuals with various cardiovascular diseases exhibited hyperuricemia, with congestive heart failure and hypertensive heart disease showing the highest rates. Low physical activity, dyslipidemia, and chronic kidney disease were also significantly associated with hyperuricemia in these study participants, suggesting a potential link between hyperuricemia and these CVD risk factors. Therefore, in the context of resource-limited settings like Ethiopia, we advocate for the integration of simple and affordable uric acid screening into routine cardiovascular care. This proactive approach, combined with targeted lifestyle interventions, offers a feasible strategy for early identification and management of hyperuricemia, potentially mitigating its adverse effects on cardiovascular outcomes. While these findings warrant

further investigation, including community-based and longitudinal studies, they provide a compelling rationale for immediate action to address hyperuricemia as a critical comorbidity in Ethiopian CVD patients.

## Supporting information

**S1 Table. This is the S1 Table for model fitness for logistic regression.**
(DOCX)

**S2 Table. This is the S2 Table for SPSS output for multiple logistic regression.**
(DOCX)

**S1 File. Excel for database.**
(XLS)

## Acknowledgments

We are thankful to all the staff in the Medical Laboratory Science Department, as well as the physicians, nurses, study participants at Salale University Comprehensive Specialized Hospital, for their collaboration in the successful achievements of this study. We would also like to acknowledge Salale University for sponsorship.

## Author contributions

**Conceptualization:** Negesse Bokona Rufe, Tolera Ambisa Lamesa, Aklilu Getachew Mamo, Bedasa Addisu, Sintayehu Asaye Biya.

**Data curation:** Belay Merkeb Zewude, Bedasa Addisu, Deresa Jamma Nigusie.

**Formal analysis:** Belay Merkeb Zewude, Bedasa Addisu, Deresa Jamma Nigusie.

**Investigation:** Aklilu Getachew Mamo.

**Methodology:** Belay Merkeb Zewude, Bedasa Addisu, Deresa Jamma Nigusie.

**Project administration:** Tolera Ambisa Lamesa, Sintayehu Asaye Biya.

**Supervision:** Tolera Ambisa Lamesa, Aklilu Getachew Mamo, Sintayehu Asaye Biya.

**Validation:** Negesse Bokona Rufe, Bedasa Addisu.

**Visualization:** Negesse Bokona Rufe.

**Writing – original draft:** Negesse Bokona Rufe.

**Writing – review & editing:** Negesse Bokona Rufe, Bedasa Addisu, Deresa Jamma Nigusie, Sintayehu Asaye Biya.

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
