## [Decision Letter · Decision Letter 0]

Mar 21 2025

PONE-D-24-57933Hyperuricemia and associated factors among adult cardiovascular disease patients at Salale University comprehensive specialized hospital, Fitche, Central EthiopiaPLOS ONE

Dear Dr. Rufe,

Thank you for submitting your manuscript to PLOS ONE. After careful consideration, we feel that it has merit but does not fully meet PLOS ONE’s publication criteria as it currently stands. Therefore, we invite you to submit a revised version of the manuscript that addresses the points raised during the review process. Please address in detail all issues raised by the reviewers.

We look forward to receiving your revised manuscript.

Kind regards,

Paolo Magni

Academic Editor

PLOS ONE

Journal Requirements:

2. Thank you for stating the following financial disclosure: "salale university for 25000 ethiopian birr"

Please include this amended Role of Funder statement in your cover letter; we will change the online submission form on your behalf."

Additional Editor Comments:

The paper shows several methodological flaws.The Authors are invited to revise it thorougly addressing all comments by the reviewers.

Reviewers' comments:

Reviewer's Responses to Questions

**Comments to the Author**

1. Is the manuscript technically sound, and do the data support the conclusions?

Reviewer #1: Partly

Reviewer #2: No

2. Has the statistical analysis been performed appropriately and rigorously? 

Reviewer #1: No

Reviewer #2: No

3. Have the authors made all data underlying the findings in their manuscript fully available?

Reviewer #1: Yes

Reviewer #2: Yes

4. Is the manuscript presented in an intelligible fashion and written in standard English?

Reviewer #1: Yes

Reviewer #2: Yes

5. Review Comments to the Author

Reviewer #1: Page 19. I believe the words 'vulvar' and 'rheumatoid' are spelt wrongly and the correct spelling should be 'valvular' and 'rheumatic'.

I could not find a definition of hypertensive heart disease, congestive heart failure, valvular heart disease, cardiomyopathy or rheumatic heart disease. How would a patient with cardiomyopathy and heart failure be classified - as heart failure or cardiomyopathy? Also how would a patient with rheumatic valve disease be classified - as valvular heart disease or rheumatic heart disease? Most rheumatic heart disease patients would have valvular heart disease.

I note that there are no patients with ischemic heart disease/coronary artery disease in the study population. Is this condition rare in the study population?

There is discrepancy in number of patients with valvular heart disease and cardiomyopathy in the text and in Table 2. Which is correct? Is the percentage of patients with valvular heart disease and cardiomyopathy 9.4% and 8.7% respectively or the other way round?

There is an error in calculation of percentages of hyperuricemia in subgroups. The percentages should be the number of patients with hyperuricemia divided by the total number of men/women by each subgroup x 100%.

Male 53/124 or 42.7% (hyperuricemia) 71/124 or 57.3% (normouricemia).

Fig 2 is referred to in the text but placed at the end of the article. I suspect that the percentages of patients displayed is divided by the wrong base (total no of patients in the study population) and not divided by the total of each subgroup. It may be better to put the actual number of patients with these conditions who have hyperuricemia and normouricemia as in Table 3. By my calculations, out of the subgroup of CHF patients: 48% have hyperuricemia and out of the HHD patients: (15.8/(15.8+22.5) = 41.2% have hyperuricemia.

I suggest recalculate the percentages in subgroups. It should not be much lower than the overall prevalence of hyperuricemia of 41.3%.

Reviewer #2: The manuscript titled "Hyperuricemia and Associated Factors Among Adult Cardiovascular Disease Patients at Salale University Comprehensive Specialized Hospital, Fitche, Central Ethiopia" aims to assess the prevalence of hyperuricemia and its associated factors among cardiovascular disease (CVD) patients in a hospital-based setting. While the study addresses an important public health issue, several methodological limitations and issues with data interpretation significantly weaken its conclusions.

General Comments:

Strengths:

The study contributes novel data on hyperuricemia prevalence in an Ethiopian population, which is underrepresented in the literature.

The focus on an important and emerging cardiovascular risk factor is relevant to the broader field of CVD research.

The use of standardized biochemical measurements strengthens the reliability of the data.

Weaknesses:

The cross-sectional design limits the ability to infer causality between hyperuricemia and CVD.

The study does not sufficiently discuss the role of dietary patterns and genetic predisposition, both of which are crucial factors in hyperuricemia development.

The logistic regression model appears to lack key adjustments for potential confounders such as medication use (e.g., diuretics, allopurinol) and socioeconomic status.

The rationale for selecting some variables in the multivariable model is unclear, and several reported associations may be confounded.

The discussion overinterprets findings and does not sufficiently compare results with existing studies from other regions.

Detailed Comments:

Introduction:

The introduction provides sufficient background on hyperuricemia and its link to CVD. However, it lacks a critical discussion on why this study is needed in an Ethiopian context beyond the absence of prior studies.

The authors should clarify whether previous research in Sub-Saharan Africa has suggested a unique risk profile for hyperuricemia in CVD patients.

Methods:

The sampling method (consecutive sampling) may introduce selection bias, as patients with more severe disease or those frequently seeking care are more likely to be included.

The study does not clarify whether participants were fasting before blood sample collection, which is essential for accurate biochemical analysis.

There is no justification for why a hospital-based population is appropriate for estimating hyperuricemia prevalence, given that hospital samples often overrepresent sicker individuals.

The statistical methods need refinement; the criteria for including variables in multivariate analysis should be explicitly stated, and a stepwise approach should be justified.

The lack of dietary assessment is a major limitation since diet is a well-established determinant of uric acid levels.

Results:

The prevalence of hyperuricemia (41.3%) is high, but the study does not explain whether this figure aligns with prior epidemiological data from similar populations.

There is no stratified analysis by gender or age groups beyond univariate associations, which could provide more insights.

The observed association between low physical activity and hyperuricemia (AOR: 4.1, p=0.004) should be interpreted with caution, as residual confounding is likely.

The association between hyperuricemia and dyslipidemia (AOR: 2.7, p=0.01) is plausible, but the study does not clarify whether this is independent of obesity or metabolic syndrome.

The authors should include model diagnostics (e.g., goodness-of-fit tests) to confirm the appropriateness of their regression analysis.

Discussion:

The discussion should better contextualize the findings by comparing them with studies from similar populations (e.g., other African nations or low-income settings).

The authors should critically assess whether hyperuricemia is a causal risk factor for CVD in their population or simply a marker of other underlying metabolic disturbances.

There is little discussion on the potential impact of renal function beyond stating an association with chronic kidney disease (CKD). The role of uric acid clearance should be explored further.

The manuscript would benefit from a discussion of public health implications, particularly regarding screening and management strategies for hyperuricemia in Ethiopia.

The limitations section should explicitly mention selection bias, lack of dietary assessment, and the inability to establish temporality in associations.

Conclusion:

The conclusion overstates the study’s findings and policy implications. The claim that early diagnosis and management of hyperuricemia are "essential" for CVD patients lacks supporting evidence from interventional studies.

The manuscript should conclude with a call for further longitudinal research rather than definitive recommendations based on cross-sectional data.

6. PLOS authors have the option to publish the peer review history of their article (what does this mean? ). If published, this will include your full peer review and any attached files.

**Do you want your identity to be public for this peer review?** For information about this choice, including consent withdrawal, please see our Privacy Policy .

Reviewer #1: No

Reviewer #2: No

---

## [Author Response · Author response to Decision Letter 1]

26 Feb 2025

dear editors;

Thank you very much for your unreserved support and guidance, based on the interest of authors, I had ordered the list of them on manuscript as following;

Negesse Bokona Rufe 2⁕, Tolera Ambisa Lamesa1¶, Aklilu Getachew Mamo1¶, Belay Merkeb Zewude2&, Bedasa Addisu 3&, Deresa Jamma Niguise2&, Sintayehu Asaye Biya1¶,

with best regards!

---

## [Decision Letter · Decision Letter 1]

May 15 2025

PONE-D-24-57933R1Hyperuricemia and associated factors among adult cardiovascular disease patients at Salale University comprehensive specialized hospital, Fitche, Central EthiopiaPLOS ONE

Dear Dr. Rufe,

Thank you for submitting your manuscript to PLOS ONE. After careful consideration, we feel that it has merit but does not fully meet PLOS ONE’s publication criteria as it currently stands. Therefore, we invite you to submit a revised version of the manuscript that addresses the points raised during the review process.

Please consider and address in full all comments on revison1 version, done by the Reviewers.

We look forward to receiving your revised manuscript.

Kind regards,

Paolo Magni

Academic Editor

PLOS ONE

Journal Requirements:

Additional Editor Comments:

Please consider and address all the comments to Revision1 version, done by the reviewers.

Reviewers' comments:

Reviewer's Responses to Questions

**Comments to the Author**

1. If the authors have adequately addressed your comments raised in a previous round of review and you feel that this manuscript is now acceptable for publication, you may indicate that here to bypass the “Comments to the Author” section, enter your conflict of interest statement in the “Confidential to Editor” section, and submit your "Accept" recommendation.

Reviewer #1: (No Response)

Reviewer #2: All comments have been addressed

2. Is the manuscript technically sound, and do the data support the conclusions?

Reviewer #1: Partly

Reviewer #2: Yes

3. Has the statistical analysis been performed appropriately and rigorously? 

Reviewer #1: Yes

Reviewer #2: Yes

4. Have the authors made all data underlying the findings in their manuscript fully available?

Reviewer #1: Yes

Reviewer #2: Yes

5. Is the manuscript presented in an intelligible fashion and written in standard English?

Reviewer #1: Yes

Reviewer #2: Yes

6. Review Comments to the Author

Reviewer #1: Abstract: Result: ..The highest prevalence of hyperuricemia was found among cardiovascular disease patients with congestive cardiac failure (16.2% - this is wrong, the correct number should be 48.0% as from Fig 2 chart) and hypertensive heart disease (15.8%, correct 41.2%).

Reviewer #2: - Minor Concerns:

Some terminologies and operational definitions (e.g., hyperuricemia, dyslipidemia, CKD) need to be more explicitly cited and consistently described across sections.

The discussion of prevalence comparisons could benefit from additional explanation of population or methodological differences that may explain discrepancies with international literature.

There is still a need to tighten the language and reduce some redundancy, particularly in the discussion and background sections.

- Detailed Comments:

Abstract:

Clearly define how hyperuricemia was measured (thresholds by sex).

Consider briefly including the hospital’s setting/context in the background sentence for international readers.

Introduction:

The background is comprehensive, but the rationale should be more focused on why studying hyperuricemia in the Ethiopian CVD population is urgent.

Remove some repetitive sentences on uric acid pathophysiology and clinical consequences.

Methods:

Please cite a specific reference for the dyslipidemia criteria from NCEP ATP III.

Mention whether quality control procedures for biochemical assays followed international guidelines (e.g., CLSI or WHO).

Results:

Results are clearly presented.

Figures and tables are useful. Ensure that all percentages are consistently reported with the same number of decimal points.

Discussion:

Briefly discussing dietary or genetic predispositions in Ethiopia that may influence uric acid levels would improve this section.

Strengthen the section on the implications for public health or clinical interventions based on your findings.

Limitations:

The limitation regarding diet and purine intake is appreciated and well discussed.

Add a note that self-reported lifestyle variables (e.g., physical activity, alcohol consumption) may be subject to recall bias.

Conclusion:

Concise and consistent with the findings. Emphasize the importance of screening and early management strategies in resource-limited settings.

7. PLOS authors have the option to publish the peer review history of their article (what does this mean? ). If published, this will include your full peer review and any attached files.

**Do you want your identity to be public for this peer review?** For information about this choice, including consent withdrawal, please see our Privacy Policy .

Reviewer #1: No

Reviewer #2: No

---

## [Author Response · Author response to Decision Letter 2]

4 May 2025

Dear Editors and reviewers: Thank you for your insightful feedback on our manuscript. We appreciate your time and expertise, which have been instrumental in strengthening the quality of our work. We have carefully addressed each of your comments and believe the revised manuscript now reflects these improvements.

with best regards!

---

## [Decision Letter · Decision Letter 2]

Hyperuricemia and associated factors among adult cardiovascular disease patients at Salale University Comprehensive Specialized Hospital, Fitche, Central Ethiopia

PONE-D-24-57933R2

Dear Dr. Negesse Bokona Rufe,

We’re pleased to inform you that your manuscript has been judged scientifically suitable for publication and will be formally accepted for publication once it meets all outstanding technical requirements.

Kind regards,

Paolo Magni

Academic Editor

PLOS ONE

Additional Editor Comments (optional):

The paper has been properly improved according to 2 rounds of revision.

Reviewers' comments:

Reviewer's Responses to Questions

**Comments to the Author**

1. If the authors have adequately addressed your comments raised in a previous round of review and you feel that this manuscript is now acceptable for publication, you may indicate that here to bypass the “Comments to the Author” section, enter your conflict of interest statement in the “Confidential to Editor” section, and submit your "Accept" recommendation.

Reviewer #1: All comments have been addressed

2. Is the manuscript technically sound, and do the data support the conclusions?

Reviewer #1: (No Response)

3. Has the statistical analysis been performed appropriately and rigorously? 

Reviewer #1: (No Response)

4. Have the authors made all data underlying the findings in their manuscript fully available?

Reviewer #1: (No Response)

5. Is the manuscript presented in an intelligible fashion and written in standard English?

Reviewer #1: (No Response)

6. Review Comments to the Author

Reviewer #1: None.

7. PLOS authors have the option to publish the peer review history of their article (what does this mean? ). If published, this will include your full peer review and any attached files.

**Do you want your identity to be public for this peer review?** For information about this choice, including consent withdrawal, please see our Privacy Policy .

Reviewer #1: No

---

## [Editor Report · Acceptance letter]

PONE-D-24-57933R2

PLOS ONE

Dear Dr. Rufe,

I'm pleased to inform you that your manuscript has been deemed suitable for publication in PLOS ONE. Congratulations! Your manuscript is now being handed over to our production team.

Kind regards,

on behalf of

Prof. Paolo Magni

Academic Editor

PLOS ONE